# Contrastive Embeddings for Neural Architectures

## Abstract

The performance of algorithms for neural architecture search strongly depends on the parametrization of the search space. We use *contrastive learning* to identify networks across different initializations based on their data Jacobians and their number of parameters, and automatically produce the first architecture embeddings independent from the parametrization of the search space. Using our contrastive embeddings, we show that traditional black-box optimization algorithms, without modification, can reach state-of-the-art performance in Neural Architecture Search. As our method provides a unified embedding space, we successfully perform *transfer learning* between search spaces. Finally, we show the evolution of embeddings during training, motivating future studies into using embeddings at different training stages to gain a deeper understanding of the networks in a search space.

## 1 Introduction

Traditionally, the design of state-of-the-art neural network architectures is informed by domain knowledge and requires a large amount of manual work to find the best hyperparameters. However, automated architecture search methods have recently achieved state-of-the-art results on tasks such as image classification, object detection, semantic segmentation and speech recognition (Ren et al., 2020).

While a significant amount of work has gone into improving the algorithms used for architecture search, there has been only limited work on constructing the space in which these algorithms operate. There exists a vast number of different ways in which the architectures in a search space can be encoded, and the effects of these choices will affect the performance of the search algorithms.

Previous embedding methods such as (Yan et al., 2020) have focused on preserving the edit distance between the computational graph of the architectures in the embedding space. Mellor et al. (2020) have showed that statistics computed on architectures at initialization, before training, can be used to infer which will perform better after training. Motivated by this, our intent is to automatically learn such statistics at initialization and generate an embedding space based on them, so that the resulting embedding space preserves fundamental properties of the architectures.

To achieve our goal, we leverage contrastive learning, that has gathered interest in the computer vision community, and produced various state-of-the-art results (He et al., 2020; Chen et al., 2020; Caron et al., 2020; Grill et al., 2020; Zbontar et al., 2021). In contrastive learning, the model learns an informative embedding space through a self-supervised pre-training phase: from the images in a batch, pairs are generated through random transformations, and the model is trained to generate similar (dissimilar) embeddings for similar (dissimilar) images.

In this work, we combine contrastive learning with the theory presented by Wang et al. (2016) on the Jacobians of networks at initialization, in order to find an embedding space suitable for Neural Architecture Search. The embedding space that we generate is invariant to the search space of origin, allowing us to accomplish transfer learning between different search spaces.

## 1.1 MOTIVATIONS AND CONTRIBUTIONS

We design a method to produce architecture embeddings using Contrastive Learning and information available from their initial state. Our technique is capable of generating embeddings that are independent from the parametrization of the search space, and evolve during training. We leverage these contrastive embeddings in Neural Architecture Search using traditional black-box optimization algorithms. Moreover, since they provide a unified embedding space across search spaces, we exploit them to perform transfer learning between search spaces.

**Parametrization-independent embeddings** NAS methods promise to outperform random search, however the encoding of the architectures must show some structure for the search algorithm to exploit. These encodings are typically produced by condensing all the parameters used to generate an architecture into a single vector. The method used to generate architectures from a search space thereby implicitly parametrizes it. The parametrization of the search space affects the performance of a NAS algorithm, as noted by e.g. White et al. (2020); Ying et al. (2019); Wang et al. (2019). However, when performing architecture search, it is not feasible to test multiple different parametrizations of the search space and evaluate which one performs better: once we have started to evaluate networks in a search space, there is no reason to discard previous evaluations. While the current generation of NAS alleviates the need for experts in the design of architectures, now expert knowledge is needed to build and parametrize a search space compatible with the chosen search algorithm, implying that it is exceedingly difficult to outperform a simple random search.

We present in Sec. 4 the first method to create networks embeddings without relying on their parametrization in any search space, through the combination of modern contrastive learning and the theory of data Jacobian matrices for neural architecture search.

**Evolution of the embeddings during training** In Section 4.4, we show how the embeddings vary during training, noting that the training procedure tends to connect areas with similar final test accuracy together. We hypothesize that this information could enable even more efficient architecture search methods in the future.

**Leveraging traditional black-box algorithms** Existing methods to generate architecture embeddings rely on metrics from their computational graphs to identify similar architectures, either by explicitly trying to preserve the edit distance in the embedding space, or by leveraging more sophisticated methods from graph representation learning. Our method leverages the information contained in the Data Jacobian Matrix of networks at initialization to train a contrastive model. As such, it can produce embeddings that more meaningfully capture the structure of the search space. As a result, traditional black-box algorithms perform well for architecture search, as shown for NAS-Bench-201 (Dong & Yang, 2020) in Section 5.1.

**Transfer learning between search spaces** Our method provides a unified embedding space, since it does not depend on the parametrization of networks in any search space. We exploit this feature to perform for the first time transfer learning between the two search spaces. In practice, we perform it between the size and the topology spaces in NATS-Bench (Dong et al., 2020) in Section 5.3.

## 2 RELATED WORK

**Neural Architecture Search** Previous works have attempted to improve network embeddings: Klyuchnikov et al. (2020) use *graph2vec* (Narayanan et al., 2017) to find embeddings such that networks with the same computational graph share the same embeddings, and similarly Yan et al. (2020) produce embeddings that are invariant to graph isomorphisms. However, the method differs in that this work trains a variational graph isomorphic autoencoder to produce the embeddings. They show that their architecture embeddings *arch2vec* perform better on downstream architecture search than supervised alternatives, additionally the embeddings that they produce approximately preserves the edit distance of the DAGs in a continuous space. Wei et al. (2020) uses a contrastive loss to find a low dimensional metric space where the graph edit distances of the original parametrization is approximately preserved. In the absence of dense sampling, all of these works rely on the prior that the edit distance is a good predictor for relative performance. In contrast, our method, learns to

find an embedding space based on intrinsic properties of the architectures. It can therefore discover properties about the architectures which are not present in their graph representation.

**Data Jacobian**   Methods based on the Jacobians with respect to the input of trained networks have been shown to provide valuable information for knowledge transfer and distillation (Czarnecki et al., 2017; Srinivas & Fleuret, 2018), as well as analysis and regularization of networks (Wang et al., 2016).

**Neural Tangent Kernel**   The Jacobian of the network with respect to the parameters is computed for inference with *neural tangent kernels* (NTK) (Jacot et al., 2018). Using NTK as a proxy for NAS (Park et al., 2020) underperforms the *neural network Gaussian process* (NNGP) kernel. The NNGP provides an inexpensive signal for predicting if an architecture exceeds median performance, but it is worse than training for a few epochs in predicting the order of the top performing architectures.

**Contrastive learning**   Different techniques have been developed in contrastive learning. He et al. (2020) train a network with a contrastive loss against a memory bank of negative samples produced by a slowly moving average version of itself. Chen et al. (2020) remove the memory bank and just consider negative samples from within the same minibatch. Grill et al. (2020) remove the negative samples completely but stabilize the training by encoding the positive samples using a momentum encoder. Zbontar et al. (2021) use the correlation matrix between the features of the different augmentations and maximize the similarity of the same feature while minimizing the redundancy between features.

## 3   BACKGROUND

### 3.1   TRADITIONAL ARCHITECTURE EMBEDDINGS

A decision tree is created either implicitly or explicitly to sample networks from a search space. To encode an architecture, one records all choices as the decision tree is traversed into a vector, which is then used as the embedding of the architecture. Without any additional knowledge, a NAS algorithm will assume that all choices in the decision tree have an equal importance on the characteristics of an architecture. One commonly used encoding scheme consists of choosing the operations on nodes along with a binary adjacency matrix connecting them.

### 3.2   DATA JACOBIAN

Extended Data Jacobian matrices are used by Wang et al. (2016) to analyze trained neural networks. We ground our work in their theoretical setting, and introduce the relevant concepts below.

Multi Layer Perceptrons with ReLU activations are locally equivalent to a linear model: the ReLU after a linear layer can be combined into a single linear layer, where each row in the matrix is replaced by zeros if the output pre-activation is negative.

$$\text{ReLU}(Wx) = \hat{W}x, \quad \hat{W}_{ij} = \begin{cases} W_{ij} & \text{if } (Wx)_j \geq 0 \\ 0 & \text{otherwise} \end{cases}$$

Since matrix multiplication is closed, within a neighborhood where the signs of all neurons pre-activation is constant, the full network can be replaced by a single matrix. This property can be extended to any model whose layers can be rewritten as linear layers, including convolutional layers and average pooling layers. Max pooling layers also retain this property, and can be treated similarly to ReLU.

Therefore, in a local neighbourhood close to $x$, the full information of a network, $f$, is contained within its *Data Jacobian Matrix* (DJM).

$$DJM(x) = \frac{\partial f(x)}{\partial x}$$

and within that neighbourhood

$$f(x) = DJM(x)x$$

We can evaluate the Data Jacobian Matrix at many different points $x$ to gather information about multiple different neighbourhoods. If we assume the network to have a single output, its DJM is a vector, and we can then stack the DJMs at different points to form the *Extended Data Jacobian Matrix* (EDJM). If a network has multiple outputs we can sum them to get a single output, which we use to calculate the EDJM.

Wang et al. (2016) use the singular values of the EDJM to compute a score, and empirically show that the score is correlated with the depth of the network, and its model capacity.

### 3.3 CONTRASTIVE LEARNING

Contrastive learning is a self-supervised method that finds an informative embedding space of the input data useful for downstream tasks. Central to contrastive learning is the concept of a *view* of an object: two different views of the same object are only superficially different, and we should be able to train a network to see past these differences and identify the same underlying object. To this end, a network is trained to map different views of the same object close to each other in the embedding space and, conversely, views of different objects should be far apart from each other.

## 4 CONTRASTIVE EMBEDDINGS FOR NEURAL ARCHITECTURES

We leverage intrinsic properties of the networks to encode them without depending on their parametrization. We must rely on properties of the architectures at initialization, since it is not computationally feasible to train the architectures to obtain their embeddings. At variance with previous work, we develop a method to find such properties automatically, using contrastive learning.

To this effect, we train a network that takes an architecture at initialization as input and produces an embedding at the output. It is desirable that the network has the following two properties:

- Different initializations of the same architecture should yield similar embeddings.
- Different architectures should yield different embeddings.

We can therefore frame our embedding problem as a contrastive learning task: different initializations of the same network will correspond to different views of the same sample in the contrastive learning framework.

### 4.1 OUR METHOD

We compute the Extended Data Jacobian Matrix (EDJM) of architectures at initialization, we rescale each weight matrix by a random and log-uniform scalar in the range $(\frac{7}{8}, \frac{8}{7})$ and we use a low-rank projection of it as input to our contrastive network, to limit memory requirements. We will refer to the projected version of the EDJM as the *Extended Projected Data Jacobian Matrix* (EPDJM).

$$\text{EPDJM}(X)_i = \phi_X \left( \frac{\partial \|f(X_i)\|_1}{\partial X_i} \right) \tag{1}$$

where $\phi_X$ denotes a projection onto the top-k principal components.

$$X = \begin{bmatrix} U_1 & U_2 \end{bmatrix} \begin{bmatrix} \Sigma_1 & 0 \\ 0 & \Sigma_2 \end{bmatrix} V^T, \quad \phi_X(x) = U_1 \Sigma_1 x \tag{2}$$

The contrastive network is then applied to the EPDJM, and trained using Barlow Twins (Zbontar et al., 2021). Once the Contrastive Network is trained, we can obtain an embeddings of any architecture in the search space, as shown in Figure 1. The embeddings produced by the contrastive network is high dimensional and non-uniform which may be detrimental for some downstream methods. In order to solve both of these problems, we seek to find a low dimensional space where the distances

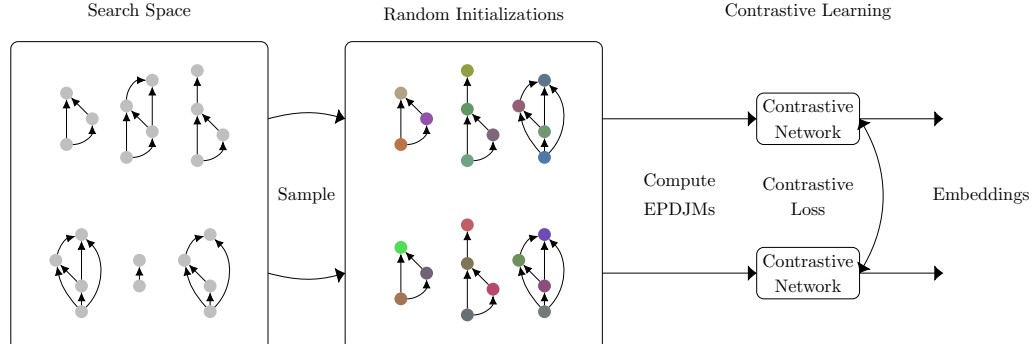

Figure 1: Illustration of our method for obtaining the network embeddings. We sample architectures from the search space, and form a batch of views with different random initializations. We compute the data Jacobians, project them, and feed them to the contrastive network. The contrastive model learns to generate similar embeddings for networks with similar performance.

between the embeddings is approximately conserved and the volume associated with each architecture is approximately uniform. Formally, we would like to find a low-dimensional embedding that minimizes the Gromov-Wasserstein distance (Mémoli, 2011):

$$\underset{\pi \in \Pi}{\arg\min} \int_{X^2 \times Y^2} |d_x(x, x') - d_y(y, y')|^2 d_\pi(x, y) d_\pi(x', y') \tag{3}$$

Where $d_x(\cdot)$ is the distance in the high dimensional embedding space, $d_y(\cdot)$ is the distance in the low dimensional space, and $\Pi$ is the set of mass-conserving transport plans between the measures. Since the target measure is continuous, we have a semi-discrete Gromov-Wasserstein optimal transport problem to which we are not aware of any numerical solvers.

Instead we approximate it with the following two step processs.

$$\underset{y}{\arg\min} \sum_{i,j} |d_x(x_i, x_j) - d_y(y_i, y_j)|^2 \tag{4}$$

$$\underset{\pi \in \Pi}{\arg\min} \int_{Y \times Y} d_y(y, y')^2 d_\pi(y, y') \tag{5}$$

We solve the first problem with gradient descent. The second is a semi-discrete Wasserstein optimal transport problem, which we solve with multi-scale averaged gradient descent, as described in Leclaire & Rabin (2019). Informally, the first step finds a low-dimensional space where the distances between the embeddings are approximately preserved, whereas the second step makes sure that the volume associated with each architecture is uniform. We use the cosine distance in the source space and the Euclidean distance in the low dimensional target space.

For simplicity, for the downstream methods we use the centroids of the resulting Laguerre cells for the final embeddings. To facilitate interpretability we use a two-dimensional embedding space.

Finally, we notice that the resulting embedding space does not accurately capture the number of parameters of the architectures which is an important predictor of the final accuracy (Hestness et al. (2017); Kaplan et al. (2020)), therefore we compliment our embeddings with a third dimension containing the logarithm of the number of parameters of the architecture.

### 4.2 IMPLEMENTATION

Since the input to the EDPJMs are two dimensional and contains structure both in the Jacobian dimension and the data-point dimension, we chose to use an MLPMixer architecture (Tolstikhin

et al., 2021) for the contrastive network, which alternates between applying transformations over the different dimensions.

For the contrastive learning, we use Barlow Twins (Zbontar et al., 2021) with a batch size of 512, and $\lambda = 5 \cdot 10^{-3}$ as suggested in the original paper. We project the Jacobians down to a 256-dimensional space. To accelerate the contrastive learning, we precompute the projected Jacobians using four different initializations for each architecture. The computation of the projected Jacobians takes less than 2 hours for NasBench201 on a single GPU (NVIDIA RTX 2080Ti). The embedding size is set to 512, and we use a single layer feedforward network for the projection head.

## 4.3 ANALYSIS OF THE EMBEDDINGS

We plot the t-SNE projections (Van der Maaten & Hinton, 2008) at different stages of our method in Figure 2 to analyze the influence of the contrastive learning on the embeddings. We note that the EPDJM alone carries some meaning in the t-SNE space. The contrastive embeddings at initialization of the network already exhibit more evident structure. The embeddings after the contrastive learning phase contain little noise and clearly separate architectures based on performance. Finally after using the projection based on optimal transport we have a two-dimensional embedding space where all architectures are near-uniformly laid out in the unit square.

Further, we predict the performance of the unseen networks in the search space using Random Forests (Breiman, 2001) with the default hyperparameters, to analyse the predictive power of the embeddings. The results for NAS-Bench-201 (Dong & Yang, 2020) are shown in Figure 3 and Table 1, and indicate that the contrastive embeddings are highly predictive of the performance of the architectures in this search space.

Table 1: Metrics computed on predicted accuracies for the three benchmarks in NAS-Bench-201. This provides a condensed view of Figure 3

|                | CORRELATION | KENDALL-$\tau$ |
| --- | --- | --- |
| CIFAR-10       | 0.94 | 0.73 |
| CIFAR-100      | 0.93 | 0.73 |
| IMAGENET16-128 | 0.90 | 0.68 |

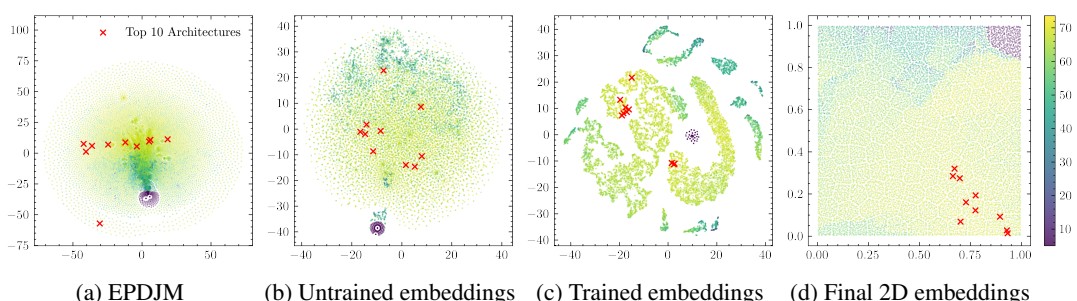

(a) EPDJM     (b) Untrained embeddings     (c) Trained embeddings     (d) Final 2D embeddings

Figure 2: t-SNE projections of different statistics of the architectures in NAS-Bench-201.

## 4.4 EMBEDDINGS DURING OPTIMIZATION

Once the contrastive network is trained, it can produce embedding for architectures at various points during their training. We show the evolution of the embeddings of 20 architectures during the 50 first epochs of training in Figure 4: the embeddings vary during the training procedure, potentially enabling future methods to learn more information about the search space for each evaluated network. In particular, the training procedure tends to connect areas with similar final test accuracy.

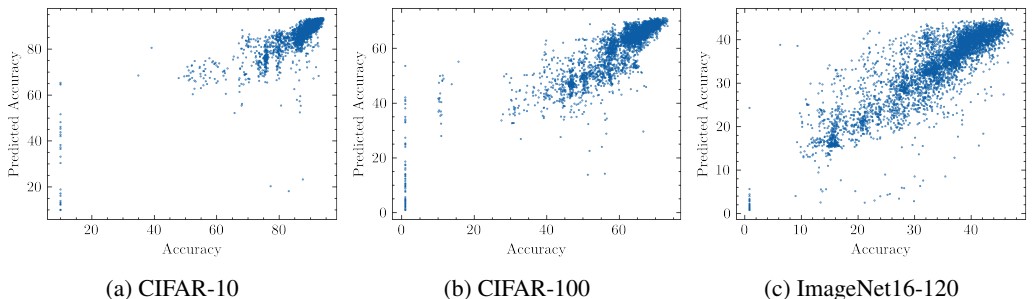

| (a) CIFAR-10 | (b) CIFAR-100 | (c) ImageNet16-120 |

Figure 3: Predicted accuracy against actual accuracy. The predictions are produced by Random Forest Regression applied on our embeddings of 1000 randomly selected architectures in NAS-Bench-201 (Dong & Yang, 2020).

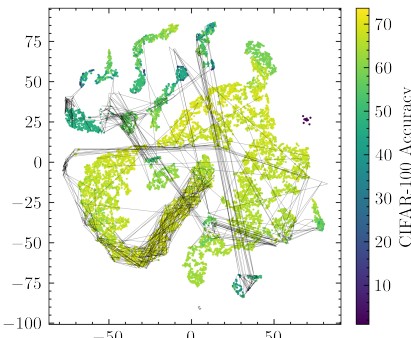

Figure 4: t-SNE projections of movement in embedding space during training of 20 architectures in the NAS-Bench-201 benchmark. The lines show the trajectories in the embedding space during training of an architecture.

## 5 ARCHITECTURE SEARCH

We evaluate our contrastive embeddings on the task of architecture search. We use two popular gradient-free optimization algorithms for the search phase, Differential Evolution, *DE*, (Storn & Price, 1997) and Tree Parzen Estimators, *TPE* (Bergstra et al., 2011). While there are countless other search algorithms, we argue that both of these algorithms should work well given a good embedding space. For both algorithms, we use the default hyper-parameters and perform no hyper-parameter tuning. We compare our embeddings, *CENA*, with *Arch2Vec* produced by Yan et al. (2020). Additionally, we compare with REINFORCE Williams (1992) using a traditional adjacency matrix encoding.

### 5.1 NAS-BENCH-201

We show the results on the NAS-Bench-201 benchmark (Dong & Yang, 2020) in Figure 5. For both downstream methods, *TPE* and *DE*, our contrastive embeddings significantly outperform the corresponding method using *Arch2Vec*. Additionally, *TPE* with our embeddings consistently outperforms REINFORCE using a categorical distribution over the possible operations at each node.

### 5.2 NAS-BENCH-101

The search space in Nas-Bench-101 is large and computing the data Jacobian matrices for all architectures would take a considerable amount of time. We compute the contrastive embeddings for a subset of $\frac{1}{25}$ of all architectures which results in $16945$ architectures. We consider the same subset when using *arch2vec* noting that using the full search space does not improve the performance.

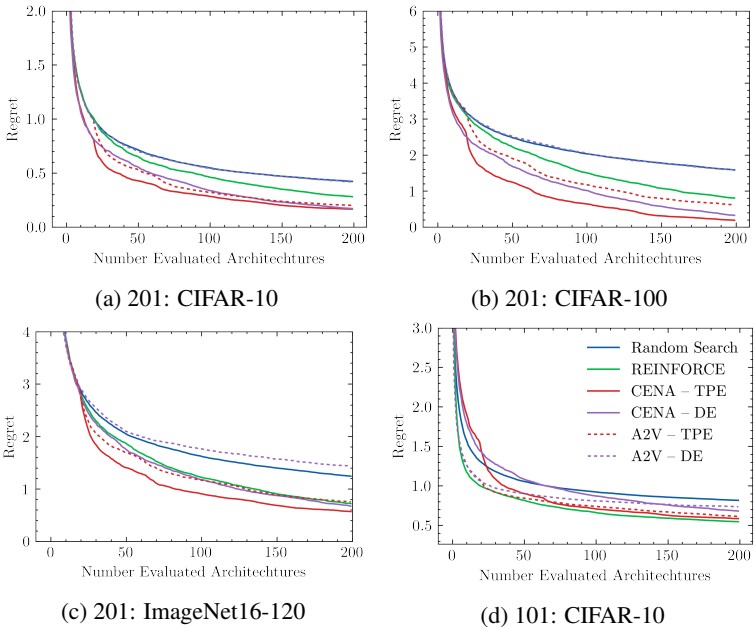

(a) 201: CIFAR-10

(b) 201: CIFAR-100

(c) 201: ImageNet16-120

(d) 101: CIFAR-10

Figure 5: Search results on NAS-Bench-201 (Dong & Yang, 2020) and NAS-Bench-101 (Ying et al., 2019)

We show the results of the search on the benchmark (Ying et al., 2019) in Figure 5d. Again, for both downstream methods our embeddings reach a lower regret at the end of the training. However, our REINFORCE implementation using a categorical distribution over the operations on the nodes as well the elements of the adjacency matrix, consistently outperforms all other methods on this benchmark.

## 5.3 Transfer Learning

A unique feature of our contrastive embeddings is that they do not depend on any information about the search space used to generated architectures, allowing us to merge the embedding spaces of multiple search spaces into a single one. With such unified embedding space, we perform transfer learning from one search space to another.

NATS-Bench (Dong et al., 2020) consists of two different search spaces: a first one (*topology*) where the topology of architectures is evaluated, and a second (*size*) where the number of filters in different convolutional layers is evaluated. We use random forests (Breiman, 2001) to predict the accuracy in one search space based on the other, and we display the results in Figure 6.

To evaluate the performance on the transfer learning task we compute the Pearson correlation co-efficient as well as the Kendall rank correlation, shown in Table 2. For both *size→topology* and *topology→size*, we see a significant correlation between predicted accuracies and the actual accuracies *without ever having evaluated a single network from the target search space*.

## 6 Conclusion and outlooks

We have developed an end-to-end method to produce embeddings for architectures, using information available from their initial state and contrastive learning, eliminating the need for manual tuning of the search space parametrization. Our analysis of the embeddings clearly shows the advantage introduced by every stage of our pipeline, and our results on Neural Architecture Search using contrastive embeddings are promising.

More precisely, by visualizing the t-SNE at different stages of our pipeline, we have shown that the embeddings produced by our technique discover the structure of the search space. We also

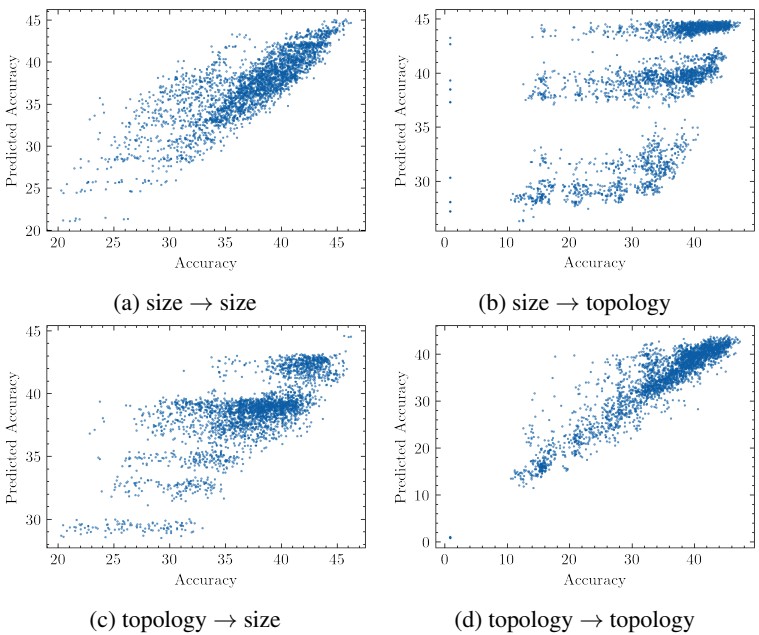

(a) size → size

(b) size → topology

(c) topology → size

(d) topology → topology

Figure 6: The transferability of features is evaluated using the two different search spaces that are provided by NATS-Bench: *size* and *topology* (Dong et al., 2020). A random forests model is trained on 5000 samples from one benchmark and evaluated on the other. The notation *size→topology* for example means that the model is trained on the size benchmark and evaluated on the topology one. For both *size→topology* and *topology→size* we see a significant correlation between the the predicted accuracies and the actual accuracies without ever having evaluated a single network from the target search space.

Table 2: Metrics computed on predicted accuracies obtained from the transfer learning. A random forest model is trained on a source search space and evaluated on the target search space. The metrics are reported as *mean ± standard deviation* aggregated over ten runs. The results from the first run can be seen in Figure 6.

| SOURCE→TARGET | CORRELATION | KENDALL-$\tau$ |
|---|---|---|
| SIZE→SIZE | $0.84 \pm 0.007$ | $0.66 \pm 0.007$ |
| SIZE→TOPOLOGY | $0.60 \pm 0.010$ | $0.51 \pm 0.014$ |
| TOPOLOGY→SIZE | $0.75 \pm 0.008$ | $0.51 \pm 0.010$ |
| TOPOLOGY→TOPOLOGY | $0.95 \pm 0.003$ | $0.74 \pm 0.003$ |

demonstrated how the embedding space evolves throughout training epochs, connecting regions of the search space with similar final performance, opening the possibility of future work to learn additional information by analyzing these trajectories. Moreover, we evaluated the performance of traditional black-box optimization algorithms using our embeddings on the task of neural architecture search and show that our embeddings perform better on the evaluated benchmarks than previously suggested embedding methods.

Finally, since the embeddings are independent of the search space, our technique provides a unified embedding space and enables the possibility to learn universal properties of the networks. We verified this by performing for the first time *transfer learning across search spaces* in neural architecture search.

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
