# OpenReview forum: "Contrastive Embeddings for Neural Architectures"
_ICLR.cc/2022/Conference — ICLR 2022 Submitted_

### Official Review · Reviewer_oXnB · 2021-11-02

**Correctness:** 3
**Technical Novelty And Significance:** 3
**Empirical Novelty And Significance:** 3
**Recommendation:** 3
**Confidence:** 3

**Main Review:**

Strengths:
+ Novel neural architecture embedding that is agnostic to topology.
+ Extensive visualizations on the learned embedding, and its effects in NAS settings. Transfer learning is a very nice experiment.

Weaknesses:
Overall I think the paper suffers from some clarity and motivation issues. I hope the authors can help clarify in the rebuttal period.

+ EDJM computation is unclear

   The exact empirical computation of EDJM is unclear:
     1. What are the sampled inputs $x$? Are they fixed? If so, are they fixed for the same architecture, or across all architectures?
     2. Why is each weight matrix only randomly scaled (instead of re-sampling each entry as usual)?

+ Second dimensionality-reduction stage motivation

   The authors motivated from (1) a dimensionality-reduction perspective and (2) wanting each architecture to have similar volume. I am not convinced by both motivations.

    For (1), the contrastively-learned embedding is 256-dimensional, which is not too high for random search or differential evolution in usual cases. Does the NAS setting require a smaller dimensionality to work? If so, can there be an ablation studies (on dimensionality, and on 2nd stage)?

   For (2), I am not following why Eqn. (5) can be used to achieve roughly uniform volume assignment. The volume here seems to refer to the volume of the set of embedding vectors. But isn't that given by Eqn. (4), which optimizes the corresponding $y$ vectors for each $x$? Also for Eqn. (4), are the $y$ vectors optimized directly or is there an encoder of some sort? If not, is the method limited to training and testing on the same fixed set of architectures? The notations here are a bit confusing. Is $d_\pi$ a differential or a distance? I was assuming the former (as an OT plan). But why is it operating on $(x, y)$ in Eqn. (3) but $(y,y')$ in Eqn. (5)?

+ Directly using EPDJM

   From Fig. 2a, the top performing architectures are already mostly lined up in EPDJM space. What if one run NAS directly on that?

+ Decoder?

   From my understanding, NAS methods like DE requires a decoding mechanism. What is the decoder here?

Minor presentation issues:
+ It'd be great to see the training time direction in Fig. 4 .

**Summary Of The Paper:**

The paper proposes a self-supervised embedding learning method to learn embeddings of various-sized neural network architectures. Each network is first represented as a low rank projection of a Jacobian matrix, where the rows are Jacobians (output-averaged if multivariate) evaluated at various inputs at random initialization time, called EDJM. Since EDJM is random, multiple such representations are treated as positive pairs for contrastive learning. From the contrastively learned embedding, a further dimensionality-reduction stage is optimized to (1) preserve distances and (2) achieve uniform volume for each architecture. The main application of the final embedding is NAS, where the method outperform baselines.

**Summary Of The Review:**

The problem of embedding various architectures into the same fixed-length vector is an interesting one. The present paper proposes an interesting approached based on contrastive-learning. However, the method is not described clearly, and some of the algorithmic choices are not well motivated. Given these considerations, I don't recommend acceptance in its current form.

---

> ### Author Response · Authors · 2021-11-14
> **Response: Review oXnB**
>
> Thank you for the review, we'll address the main points of the review in the order in which they appear.
>
> **How do we decide which inputs to use when computing the EDJMs?**
>
> We randomly sample images from the input dataset and use the same selected images for all architectures. We note that the choice of images does not affect the final performance of the algorithm significantly.
>
> **Is each weight matrix only randomly scaled?**
>
> The entirety of the network is reinitialized for each view/augmentation, however in addition we also perform rescaling of the weights. Since the weights are drawn from a random distribution with fixed scale, it can be possible for the contrastive network to identify networks based on the scaling of the EDJMs. While the rescaling is not necessary for the method to work, we nevertheless observe an improvement by using it.
>
> **Why reduce the dimensionality of the Embeddings?**
>
> While it is indeed possible to do the search using higher dimensional embeddings, a low dimensional space has its advantages.
> 1.	The volume in high dimensional spaces is rather unintuitive, if we use optimal transport to find embeddings of uniform volume it will both be costly and induce significant distortions. However, if we do not use OT the embedding space will be very non-uniform which causes black-box optimization algorithms to struggle.
> 2.	Understanding the structure of the search space is much simpler in low dimensions that are possible to visualize.
> 3.	We empirically observe higher performance in lower dimensional spaces, the search algorithm only is not able to make use of the original high dimensional space it the short search time. We note that Arch2Vec use a 16D embeddings which is similarly low dimensional though not as extreme.
> We agree that an ablation study of the performance as a function of the dimensionality of the embeddings would improve the paper.
>
> **Why is the volume associated with each architecture in the embedding space uniform?**
>
> We start by clarifying what we mean with the volume associated with an architecture. When we use a black box optimization algorithm to sample points in the embedding-space, we typically choose the architecture which is closest to the selected sample point. The probability of choosing a particular architecture is then proportional to the volume of its Voronoi-cell in the embedding space. Uniform volume of the architectures therefore means that if we’re sampling uniformly in the embedding-space we are also sampling the architectures with uniform probability. This is an appealing property as the embedding method does not know the downstream task and can therefor not predict which architectures will do well on it, it can only learns which architectures are similar and should not distort the probability of sampling a given architecture.
>
> Our approximation of the semi-continuous Gromov-Wasserstein distance is done in two parts. The first part moves each point to the lower dimensional space, which conserves its mass. There is no encoder here, we are simply finding the low dimensional points which best conserves the distances in the high dimensional space. The second part uses semi-discrete optimal transport which, by definition, is mass-conserving. In particular, the set of transport plans, $\Pi$, only contains transport plans which are mass conserving. We agree that the notation can be improved. The transport plans in (3) and (5) are operating on different spaces since the dimension reduction step has already been carried out in (4).
> Finally, you are correct in pointing out that the optimal transport step is limited to a fixed set of architectures. However, this is not a big limitation in practice, for example in NasBench-201 we compute the embeddings of all architectures at once and in NasBench-101 we randomly select 1/25 of the search space.
>
> **Can we run NAS using the raw EPDJMs?**
>
> The EPDJMs are very high dimensional, each architecture embedding would be 32768D, and they are far from uniformly positioned in the space. While possible, it would both be very slow and difficult to make use of this information in the limited search time of evaluating a few hundred architectures. While it is true that the top architectures form a line in the t-SNE space, it is a bit misleading since lines do not hold any meaning in the t-SNE space, and further the vast number of the 15k architectures lies between one end of the line and the other.
>
> **Does DE require a Decoder?**
>
> We would like to have further clarifications on this question, and in the meantime we reply to the best of our knowledge. DE is a black-box optimization algorithm, we let it freely sample the embedding space and return the accuracy of the closest architecture. The code is available in the supplementary material, (line 49-55 in make\_simulations\_201.py). We use scipy.optimize.differential\_evolution.

---

> > ### Comment · Reviewer_oXnB · 2021-11-24
> > **Reply to authors**
> >
> > Thanks for your response. It indeed cleared some of my confusing. For example, the decoding process is finding the nearest architecture.
> >
> > Unfortunately, I believe the paper still is not ready for publication, for a few reasons:
> > 1. Some (non-trivial) components of the proposed procedure is not well motivated.
> >    + The second stage (forcing uniform volume) would be much more convincing if the authors show failure results and analyses on non-uniform embeddings.
> >    + The raw EPDJMs look good, but is high-dimensional. What if we perform NAS using its t-SNE or PCA?
> > 2. Some of the limitations (while may greatly not affecting NAS results) are not clearly stated
> >    + That representation is learned wrt a fixed set of architectures may be a bit hard to realize to readers who expect an encoder of some sort (to apply to novel architectures).
> >
> > I really appreciate that the authors agree that some of the suggested experiments would strengthen the paper. I look forward to seeing an updated version in future.
> >
> > ---
> > edit: fixed formatting of the bullet list

---

### Official Review · Reviewer_pbA7 · 2021-11-03

**Correctness:** 4
**Technical Novelty And Significance:** 3
**Empirical Novelty And Significance:** 3
**Recommendation:** 5
**Confidence:** 4

**Main Review:**

Strengths:

1. This is a novel way to generate the embeddings that does not require the architecture  to be encoded into a vector representation before it is fed in.
2. It does not require computation of the accuracy.
3. They demonstrated that the downstream embeddings have high predictive power by inputing them to a random forest, which was able to predict the accuracy, size etc.

Weakness:
1. Neural Architecture Search Without Training also used jacobian matrix and show that the kendall Tau is less than 0.55. So I am curious how you are able to achieve higher performance by using the embeddings whose input is the same. I am assuming that the contrastive learning is as good as predictive power of the input
2. How would you decide which images from the dataset are used to compute the jacobian matrix? This would be crucial to the quality of the embeddings
3. The original dimension of the embeddings, 512, is not that high.Random Forest and the NAS algorithms can handle it. Why should we reduce the dimension to 2?

More Experiments requested:
1. The experiment conducted on NATS-BENCH is not enough to substantiate the claim that the embeddings can generalize across various search space. In order to bolster the claim, could you predict the accuracies for datasets in table 1 while training the contrastive network on NAS-BENCH 101 and DARTS search space and show its predictive power on NAS-BENCH 201

2. For Figure 5, Could you also include arch2vec + REINFORCE, arch2vec + Bayesian Optimization,CENA + REINFORCE and CENA + Bayesian Optimization in the plot?

3. For Table1, could you also include the performance prediction of random forest when using embeddings from NAO [1], GCN [2] and arch2vec

[1] Neural Architecture Optimization, Luo et al.
[2] Neural Predictor for Neural Architecture Search, Wen et al.



**Summary Of The Paper:**

The paper proposes a new technique to generate embeddings agnostic of the search space. This can be achieved by  first computing the data jacobian of the network with respect to datapoints sampled from different neighbourhoods. This jacobian matrix is then input to a
contrastive network which produces architecture embeddings. The contrastive views in this case are different initializations of the same network, which in turn must yield the same embedding. As these embeddings are high dimensional, they are reduced to lower dimension while ensuring that the distance between the embeddings is preserved in the lower dimensional space and the volume associated with each architecture is also preserved.

**Summary Of The Review:**

This paper is novel and have demonstrated its potential. In order to make it more compelling, they must add more empirical evaluations requested above. If they do that, I am leaning towards an accept.

---

> ### Author Response · Authors · 2021-11-14
> **Response: Review pbA7**
>
> Thank you for the review, we'll address the main points of the review in the order in which they appear.
>
> **How can we achieve better results than NAS Without Training (v1.0)?**
>
> While we use the same input to our method as the original version of NAS without training, our methods are considerably different. Their method uses a hand-crafted heuristic based on the EDJMs, whereas we train a contrastive network to produce an informative embedding space and then use random forest regression to predict the accuracies.  As they use the same heuristic for all search spaces and all downstream tasks, the performance that they can achieve is limited. We would like to remark that this is a very interesting paper, but it does not in any way limit the performance that we should be able to get.
> To be clear training a RF on the EDJMs will not reach a correlation remotely close to that of our method. The contrastive learning part of our method produces an embedding space where similar architectures are close which significantly simplifies the downstream regression task. While the information is available in the original high dimensional EDJMs, learning what information is correlated with training performance using a small set of evaluated architectures is much more difficult.
>
> **How do we decide which inputs to use when computing the EDJMs?**
>
> We randomly sample images from the input dataset and use the same selected images for all architectures. We note that the choice of images does not affect the final performance of the algorithm significantly.
>
> **Why reduce the dimensionality of the Embeddings?**
>
> While it is indeed possible to do the search using higher dimensional embeddings, a low dimensional space has its advantages.
> 1.	The volume in high dimensional spaces is rather counterintuitive, if we use optimal transport to find embeddings of uniform volume it will both be costly and induce significant distortions. However, if we do not use OT the embedding space will be very non-uniform which causes black-box optimization algorithms to struggle.
> 2.	Understanding the structure of the search space is much simpler in low dimensions that are possible to visualize.
> 3.	We empirically observe higher performance in lower dimensional spaces, the search algorithm only is not able to make use of the original high dimensional space in the short search time. We note that Arch2Vec used a 16D embeddings which is similarly low dimensional, even though not as extreme.
>
> We agree that an ablation study of the performance as a function of the dimensionality of the embeddings would improve the paper.
>
> **More experiments**
>
> To address the criticism of further experiments on transfer learning, we performed transfer learning from NasBench 101 to NasBench 201 with the following results: correlation of 0.52 +- 0.077, and Kendall tau of 0.4 +- 0.044

---

> > ### Comment · Reviewer_pbA7 · 2021-11-23
> > **Thank you for your response**
> >
> > I thank the authors for their response. It is promising to know that the method performs well in transfer learning setting. However, the authors have not provided all the experiments I requested for. While the idea is interesting, there is scope for improvement. I would like to retain my score.

---

### Official Review · Reviewer_KncE · 2021-11-07

**Correctness:** 2
**Technical Novelty And Significance:** 3
**Empirical Novelty And Significance:** 2
**Recommendation:** 5
**Confidence:** 4

**Main Review:**

The premise is interesting and the implications are exciting. The contributions in this paper are novel - to the best of my knowledge, this is the first paper that demonstrates that it may be possible to do transfer learning for NAS on related tasks.

Section 4.3 - The predicted accuracy looks quite strong. I am curious why you didn't optimize the hyper parameters of your Random Forest. Further, how was this evaluation done? I couldn't find a description of a train-test split. Random Forests can overfit on data really easily, especially in this case since you'll have only 3 features as input.

In Section 4.4, I found the premise of the experiment interesting, but I'm unable to conclude anything interesting based on Figure 4. Can you present the data in a more readable format?

5.1, 5.2 - I appreciate the simplicity of the experiment by not confounding the results by including the hyper parametrs of TPE and DE as part of the search space. However, the results in this section are mixed. It looks like CENA performs better than in the baseline in some cases, sometimes by just a small margin and sometimes even worse than REINFORCE.  Also, as was acknowledge by the authors in this section, computing Jacobians can be slow.

Apart from the final accuracy, it would be interesting to know how long each method took in wall clock time. If CENA is significantly faster than the other methods, you could make the case that it runs faster while preserving performance. On the other hand if CENA is extremely slow, then perhaps it is better to use random search to evaluate more architectures in the same time period as CENA.

5.3 - In my opinion, this is the biggest strength of the method. However, I wish more details were provided. How exactly are you'll doing transfer learning? Are you using the contrastive learning network from one dataset and using it zero-shot on the other, and then regressing on the accuracies using Random Forests? Once again, was a train-test split used for the random forests and are the reported numbers the out-of-sample performance?

**Summary Of The Paper:**

The authors use constrastive learning to embed NAS candidates into an informative search space, which is highly predictive of the architectures' actual accuracy. This embedding space allows traditional black box optimizers to perform well on NAS benchmarks.

**Summary Of The Review:**

The core idea and the benefits of this technique are exciting, however experimental details are lacking and the performance is a little lack-luster.

- More details are needed about how the Random Forests were used.

- The paper should focus more on the transfer learning aspect of this technique. I suggest you'll evaluate this technique on more datasets to build a strong case.

- The paper should list the cost of evaluating each method.

---

> ### Author Response · Authors · 2021-11-14
> **Response: Review KncE**
>
> Thank you for the review, we will address the main points of the review in order they appear.
>
> **Why not optimize the Hyperparameters of the random forests?**
>
> We believe that the performance that we show in section 4.3 is quite strong without tuning the hyper-parameters and the reviewer seems to agree. While it is likley that tuning of the hyper parameters would indeed increase the performance, we wanted to emphasize the point that it works out-of-the-box, without any tuning. We believe this to be significant for NAS as it is not possible to do any hyperparameter-tuning of the algorithms during the search phase. Once the searching has started there is no reason to discard evaluated architectures and start over.
>
> **Do we evaluate on the training set in section 4.3?**
>
> No, we do not evaluate on the training set. In the main text we state "we predict the performance of the unseen networks in the search space". Indeed, we fit the random forest on 500 randomly selected architectures and evaluate on 2000 different randomly selected architectures in the search space. We will clarify the evaluation setting in the camera-ready version. To clear any possible doubt, the embeddings are available in the supplementary material.
>
> **Could you clarify Section 4.4?**
>
> The main point we like to show in this section is that additional information is available during the training of the architectures that is not typically used in architecture search. For example, there are two disjoint areas of architectures with similar accuracy in the top and the bottom of the tSNE-plot, during training of one of the architectures its embedding changes from one to the other, showing that these two areas are similar.
> We take this point of clarity with us and will improve it in the camera-ready version.
>
> **What is the computational cost of the method?**
>
> The computational cost of the method is detailed in section 4.2, to reiterate: the generation of all 15k embeddings in NasBench201 takes about 2h on a single NVIDIA RTX 2080Ti.
> Regarding the cost of the search in wall clock time, we note that we are using the same downstream search algorithms for both embeddings, while training with CENA is faster due to the lower dimensionality of the embeddings, the search time is dominated by the evaluation of the networks, and not by the search algorithm.
>
> **Is the performance of the search good?**
>
> We believe that it is, we note that our embeddings are at the very least competitive with previous embeddings. The fact that this is possible without using any information about the search space or the structure of the architectures is quite remarkable. Our methods learns to find similar networks looking only at the Jacobians of the networks at initialization. This is a considerably harder problem, but once generated it comes with some advantages: it is not affected by how we parameterize the search space, it enables us to do transfer learning between different search spaces, and to look at the evolution of the embeddings during training.
>
> **How are we doing transfer learning?**
>
> We generate the EPDJMs from both search spaces and then apply our method to the joint dataset of EPDJMs, generating a joint embedding space for both methods.
> We then train a RF classifier on one of the search spaces and evaluate it, in a zero-shot fashion on the other. Again, we of course do not train and evaluate the RF on the same architectures. In fact, since the training and evaluation set comes from different search spaces, such an error is not possible.

---

> > ### Comment · Reviewer_KncE · 2021-11-29
> > **Response**
> >
> > Thank you for your detailed response. I now understand better why the hyperparameters of the random forest were not tuned, however, I am still curious of what could happen if we allow tuning of these parameters since this is what most practioners would do. Could have been an interesting section in the supplementary material. Also, thanks for clarifying how the evaluation in sec 4.3 was performed.
> >
> > I believe that the direction of this work is promising and I encourage the authors to conduct more experiments on the transfer learning aspect of the technique to build a strong case.

---

### Author Response · Authors · 2021-11-14
**General Response**

We thank all the reviewers for their contributions. We are glad that the reviewers highlighted the fact that our method is agnostic to topology, and that it allows to perform transfer learning across search spaces, as we believe these are the main contributions of our submission. We believe that further work on the contrastive network and its training could enable significant advances in the NAS field.

---

### Decision · Program_Chairs · 2022-01-20

**Decision:**

Reject

**Comment:**

All reviewers unanimously recommending rejecting this submission and I concur with that recommendation. However, many reviewers were quite pleased with the premise and basic concept of the submission and would have liked to see a clearer version with a bit more in terms of experiments.

I agree with the submission that the most interesting architecture search research is about the search space, not the search algorithm.
The submission uses measurements of the data Jacobian matrix at different points to construct an extended data Jacobian matrix that then is projected and serves as input to a contrastive embedding learning algorithm. The resulting architecture embeddings can be used for many different things, including architecture search.

Ultimately, I am recommending rejecting this submission not because of one single overriding weakness, but because the totality of issues the reviewers raised make it clear the submission is not strong enough to publish in its current form. I encourage the authors to continue this line of work and produce a stronger submission in the future to ICLR or another venue.